# Assessing the Influence of Operational Variables on Process Performance in Metallurgical Plants by Use of Shapley Value Regression

Xiu Liu [1] and Chris Aldrich [1,2,*]

1 Western Australian School of Mines, Minerals, Energy and Chemical Engineering, Curtin University, Bentley, WA 6845, Australia
2 Department of Process Engineering, Stellenbosch University, Private Bag X1, Stellenbosch 7602, South Africa
* Correspondence: chris.aldrich@curtin.edu.au; Tel.: +61-8-9266-4349

**Abstract:** Shapley value regression with machine learning models has recently emerged as an axiomatic approach to the development of diagnostic models. However, when large numbers of predictor variables have to be considered, these methods become infeasible, owing to the inhibitive computational cost. In this paper, an approximate Shapley value approach with random forests is compared with a full Shapley model, as well as other methods used in variable importance analysis. Three case studies are considered, namely one based on simulated data, a model predicting throughput in a calcium carbide furnace as a function of operating variables, and a case study related to energy consumption in a steel plant. The approximately Shapley approach achieved results very similar to those achieved with the full Shapley approach but at a fraction of the computational cost. Moreover, although the variable importance measures considered in this study consistently identified the most influential predictors in the case studies, they yielded different results when fewer influential predictors were considered, and none of the variable importance measures performed better than the other measures across all three case studies.

**Keywords:** Shapley value regression; variable importance analysis; calcium carbide; steel production; Boruta algorithm



## 1. Introduction

As in many other technical disciplines, the development and implementation of models are key to efficient operation of metallurgical processes. These models are often data-based, as developing first-principles models may either be too costly or infeasible. With the increasing availability of large quantities of high-quality process data, such models may provide reliable prediction of key performance indicators in process systems, as evidenced by recent examples of material characterization [1,2], process operations [3–5], and design [6]. However, if these processes are highly nonlinear and complex, as is often the case, the models may not be easily interpretable, which may be a problem if the trustworthiness of models is critical or if such models are used for diagnostic purposes.

Variable importance analysis [7] comprises a variety of methods that are used to interpret data-based models. Although most approaches are empirical, Shapley value analysis has recently attracted attention as an axiomatic approach that can be used with machine learning models [8,9].

Although the underlying principle of Shapley value regression was formulated in the 1950s [10], it has only recently been applied in conjunction with machine learning models, such as random forests [8]. In principle, regression is interpreted as a cooperative game in which the predictors are players and the model output is the reward that has to be allocated fairly among the predictors [11]. Shapley values are uniquely characterized by the following axioms [9], as informally summarized below:

- Efficiency: The worth (e.g., the target variable variance explained) of the full model is losslessly distributed among the predictors;
- Null Player: Predictors not contributing to the model are identified by Shapley values of zero;
- Symmetry: Predictors contributing equally to the model have equal Shapley values; and
- Additivity: If the model score is derived from the sum of two intermediate values, then the overall contribution allocated to a specific predictor is equal to the sum of the contributions of the predictor to each of the intermediate values.

Despite these intuitively appealing characteristics of Shapley value methods, they are not necessarily suitable or optimal feature selectors [9,12]. Relatively few comparative analyses of variable importance measures have been conducted; therefore, in this paper, we explore the application of Shapley regression analysis with random forest models. Furthermore, we propose an approximation of the approach that does not suffer from the computational cost of Shapley methods, yielding results similar to those obtained with full Shapley models but at a fraction of the computational cost.

The remainder of this paper is organized as follows. In Section 2, the overall analytical methodology is briefly described. In Sections 3–5, variable importance measures associated with random forests are investigated. In Section 6, the results are discussed. Finally, in Section 7, we summarizes the conclusions of the study.

## 2. Analytical Methodology

Random forest models were constructed, and four variable importance measures based on these models were used to evaluate the influence of different predictors on the performance of models based on simulated data, data from a calcium carbide furnace, and data from a steel plant. These measures were based on the permutation importance, an impurity criterion, Shapley regression values, and an estimate of the Shapley values based on a subset of the coalitions of predictors used in the models. These approaches are formally summarized below.

### 2.1. Random Forests

Random forests [13] are ensembles of classifications of regression trees that have become widely established in industrial processes, manufacturing, the healthcare industry, business, and finance since their development in the 1990s. These analytical tools have cutting edge capability in terms of classification and regression; can handle categorical or continuous variables, missing data [14], and high-dimension, low-sample-size problems [15]; and can be used in unsupervised learning applications [16]. In metal processing, they are used not only for their predictive capabilities but also as diagnostic tools to better understand the influence or contributions of operational or system variables on target variables [17–19].

Given a training data set ($X \in \mathbb{R}^{n \times \mathcal{M}}$) with labels ($y \in \mathbb{R}^n$) consisting of $n$ samples of $\mathcal{M}$ variables, random forests are constructed by:

i. Selecting $n_{tree}$ samples from $X$;
ii. Selecting $m_{try}$ predictors from $X$ at each split to construct a classification or regression tree;
iii. Repetition of (i) and (ii) until each terminal node of the tree has reached the maximum depth of the tree ($l_{max}$) as specified by the user or fewer, and each terminal node contains the minimum number of samples ($n_{node}$) as specified by the user;
iv. Repetition of (i), (ii), and (iii) until the specified number of trees ($K$) has been grown; and
v. Aggregation of the information from the $K$ trees in the forest averaged for regression or making use of majority voting for classification.

### 2.2. Permutation Variable Importance Measure (PVIM)

The permutation variable importance measure is based on the use of out-of-bag (OOB) data. More formally, let $\mathbf{T}_k$ be the OOB data consisting of $n_{OOB}$ samples seen by the $k'$th tree in the random forest during training, i.e.,

$$\mathbf{T}_k = \left\{ \left( \mathbf{x}_j^{(k)}, y_j^{(k)} \right) \right\}, \ k = 1, \ 2, \ \dots \ n_{tree} \text{ and } j = 1, \ 2, \ \dots \ n_{OOB} \tag{1}$$

The mean squared error (MSE) of the $k'$th tree can then be defined as:

$$MSE_k = \frac{1}{n_{OOB}} \sum_{j=1}^{n_{OOB}} \left( y_j^{(k)} - \hat{y}_j^{(k)} \right)^2 \tag{2}$$

and

$$MSE_{k,i} = \frac{1}{n_{OOB}} \sum_{j=1}^{n_{OOB}} \left( y_j^{(k)} - \hat{y}_{j \backslash \{i\}}^{(k)} \right)^2 \tag{3}$$

where $\hat{y}_j^{(k)}$ and $\hat{y}_{j \backslash \{i\}}^{(k)}$ are the predictions of the model before and after random permutation of the $i'$th variable $X_i$, respectively.

The permutation variable importance measure ($PVIM_i^{(k)}$) of the $k'$th tree for the $i'$th variable is then:

$$PVIM_i^{(k)} = MSE_{k,i} - MSE_k \tag{4}$$

and

$$PVIM_i = \frac{1}{n_{tree}} \sum_{k=1}^{n_{tree}} PVIM_i^{(k)} \tag{5}$$

For categorical output, $PVIM_i$ is defined as the mean difference between the error rates of the OOB data after and prior to permutation of the values of variable $X_i$.

### 2.3. Impurity Variable Importance Measure (IVIM)

Unlike the permutation variable importance measure, which is model-agnostic, impurity variable importance measures, such as the Gini variable importance measure [20] and cross-entropy measure are associated with the structure of the random forest model. At each node in each tree of the forest, the selection of both the variable to split and the split point is based on maximizing the decrease in the impurity index of the node.

More formally, considering a binary classification problem, let $p_F(C_i)$ be the frequency or fraction of the samples allocated to class $C_i$ $i = 1, 2$. The Gini index of the node can then be defined as:

$$GI_F = \sum_{i \neq j}^{C} p_F(C_i) p_F(C_j) = 1 - \sum_{i \neq j}^{C} p_F^2(C_i) \tag{6}$$

The change in the Gini index that is maximized when the node is split is:

$$\Delta \mathrm{GI} = GI_F - p_R GI_R - p_L GI_L \tag{7}$$

where $GI_L$ and $GI_R$ are the Gini indices of the left and right descendent node, respectively, in the binary split; and $p_L$ and $p_R$ are the fractions of samples associated with the left and right descendent node, respectively.

$GVIM_i^{(k)}$ of variable $X_i$ in the $k'$th tree is defined as the sum of the decreases in the impurity indices of the nodes splitting on variable $X_i$. $GVIM_i$ of variable $X_i$ is determined by summing or averaging $GVIM_i^{(k)}$ across all the trees in the random forest.

This approach is readily extendable to regression trees [16], for which the mean square error is typically used as the impurity measure; that is, splitting of nodes in regression trees

is not based on the Gini index as such, but the prediction errors in the descendent nodes are minimized, i.e.,

$$II_F = \sum_{\mathbf{X}_{i,L}} (y_i - \hat{y}_i)^2 + \sum_{\mathbf{X}_{i,R}} (y_i - \hat{y}_i)^2 \tag{8}$$

The importance of a predictor in a random forest model used for regression is essentially based on the weighted average of the incremental purities associated with each variable (split) using the node population as a weight. In this paper, these variable importance measures are referred to as impurity variable importance measures (IVIMs), with $IVIM_i$ indicating the impurity variable importance of the $i'$th of $\mathcal{M}$ predictors.

### 2.4. Shapley Variable Importance Measure (SVIM)

Consider a set $(S)$ of $\mathcal{M}$ predictors with a reward function $(v : P(\mathcal{M}) \to R_v)$, such that $v(\varnothing) = 0$. $R_v \subseteq \mathbb{R}$ is a real number, and $P(\mathcal{M})$ is a family or coalition of sets over $\mathcal{M}$. If $S \subset \mathcal{M}$, then $v(S)$ is the reward generated by the coalition $S$ when they cooperate. The Shapley value of the $i'th$ predictor in the set is defined by Equation (9), where $|S|$ is the cardinal number of $S$, i.e., the number of predictors in the coalition.

$$\phi_i(v, \mathcal{M}) = \sum_{S \subseteq \mathcal{M} \setminus \{i\}}^{\mathcal{M}} \frac{(\mathcal{M} - |S| - 1)!S!}{\mathcal{M}!} [v(S \cup \{i\}) - v(S)] \tag{9}$$

In this investigation, the reward function $(v)$ is defined as the coefficient of determination of the model, i.e., $v = SVIM_i = R_i^2$.

### 2.5. Approximate Shapley Variable Importance Measure (ASVIM)

Generating Shapley values for the predictors is an non-deterministic polynomial-time (NP)-hard optimization problem with an exponential computational time related to $2^{\mathcal{M}}$. In practice, this means computation of Shapley values for high-dimensionality systems is not feasible.

Therefore, instead of considering all the coalitions to generate the Shapley values, only the coalitions with the largest weights are considered. As indicated in Figure 1, these are essentially the models containing either very small or very large coalitions of predictors. More specifically, only the $2\mathcal{M}$ coalitions defined by $S = \varnothing$, $S = \{i\}$, $S \subset \mathcal{M} \setminus \{i\}$ and $S = \mathcal{M}$, for $i = 1, 2, \ldots \mathcal{M}$ are considered.

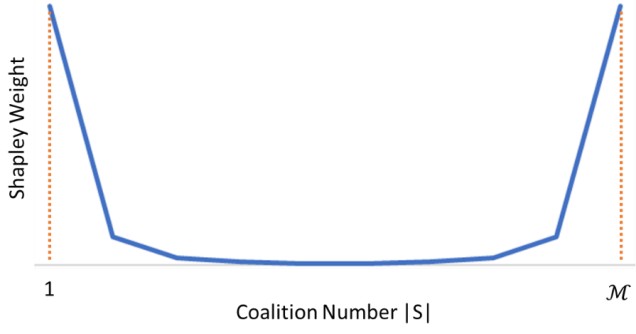

**Figure 1.** Typical U-shaped distribution of Shapley weights of coalitions of predictors.

This reduces the time complexity to a linear relationship, i.e., related to $2\mathcal{M}$.

As before, the variable importance measure is defined by $v = SVIM_i^* = R_i^2$, with the '*' superscript indicating an estimate based on the reduced set of coalitions considered.

### 3. Case Study 1: Introductory Example with Simulated Data

#### 3.1. Simulated Data with No Correlation between Variables

A simulated data set with 1000 samples was generated, where $X \in \mathbb{R}^{1000 \times 3}$ and $Y \in \mathbb{R}^{1000 \times 1}$, where $Y = X_2^3 + \frac{1}{X_3}$. The matrix $(X)$ consisted of multivariate random

numbers with a mean vector of $\overline{\mathbf{x}} = [0\ 0\ 0]$ and a covariance matrix ($\mathbf{\Sigma}$). As indicated by Equation (10), the three variables have no correlation with each other.

$$\sum = \begin{bmatrix} 1 & 0 & 0 \\ 0 & 1 & 0 \\ 0 & 0 & 1 \end{bmatrix} \tag{10}$$

Random forest models were fitted to the data, with the following optimal hyperparameters: $K = 100$; $m_{try} = 3$; $n_{try} = 80\%$ of the data; minimum leaf size, $n_{leaf} = 5$. On average, the random forest model could explain 67% of the variance of the response variable. For the SVIM, and ASVIM, the data were randomly divided into training and test sets with a ratio of 80 to 20, respectively.

The results of the variable importance analysis are shown in Figure 2. All four variable importance measures (PVIM, IVIM, SVIM, and ASVIM) identified $X_3$ as the most important variable. PVIM and IVIM identified $X_2$ as the second most important and $X_1$ as the least important. In contrast, SVIM and ASVIM could not distinguish between the importance of $X_1$ and $X_2$, as indicated by the fact that the median values of these indicators are within the ranges of the notches of the box plots. Moreover, SVIM indicated that both these variables contribute essentially zero to the variance explained by the model. The same results would likewise be obtained with rescaled ASVIM values.

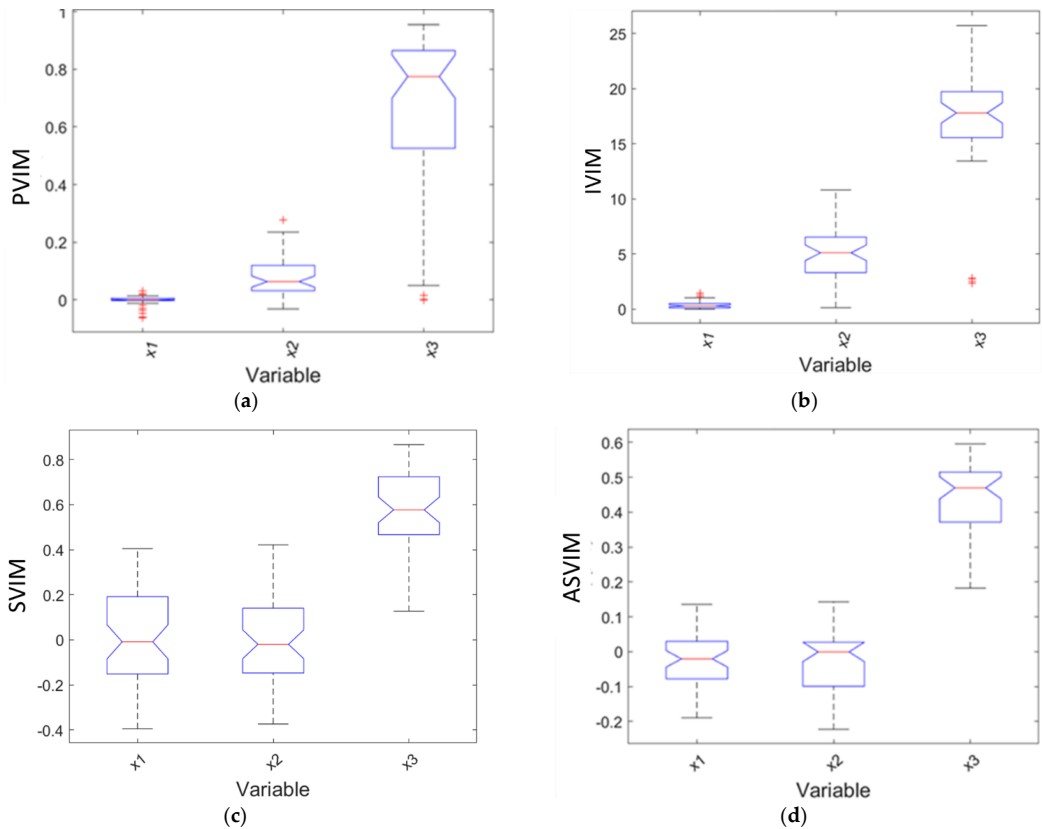

**Figure 2.** Relative importance of the uncorrelated predictors in Case Study 1 based on 50 runs, showing (**a**) permutation indices, MSE on OOB data, (**b**) impurity indices on OOB data, (**c**) Shapley values, $R^2$ on test data, and (**d**) approximate Shapley values ($R^2$) on test data.

### 3.2. Simulated Data with Strong Correlation between Variables

A simulated data set with 1000 samples was generated, where $X \in \mathbb{R}^{1000 \times 3}$ and $Y \in \mathbb{R}^{1000 \times 1}$, where $Y = X_2^3 + \frac{1}{X_3}$. The matrix ($X$) comprised multivariate random numbers with a mean vector of $\overline{\mathbf{x}} = [0\ 0\ 0]$ and a covariance matrix ($\mathbf{\Sigma}$). As indicated by Equation (11),

the first and second variables were strongly correlated with each other, whereas the third variable was independent of the other two variables.

$$\Sigma = \begin{bmatrix} 1 & 0.9 & 0 \\ 0.9 & 1 & 0 \\ 0 & 0 & 1 \end{bmatrix} \tag{11}$$

Random forest models were fitted to the data, with the following optimal parameters: $K = 100$; $m_{try} = 2$; $n_{try} = 80\%$ of the data; minimum leaf size, $n_{leaf} = 5$. On average, the random forest model could explain 70% of the variance of the response variable. As before, for SVIM and ASVIM, the data were randomly divided into training and test sets with a ratio of 80 to 20, respectively.

The results obtained with the variable importance measures are shown in Figure 3. The ability of both permutation and impurity variable importance measures to discriminate between predictors were adversely affected with increased correlation between predictors [14] With the exception of PVIM, none of the variable importance measures could discriminate between the effects of variables $X_1$ and $X_2$.

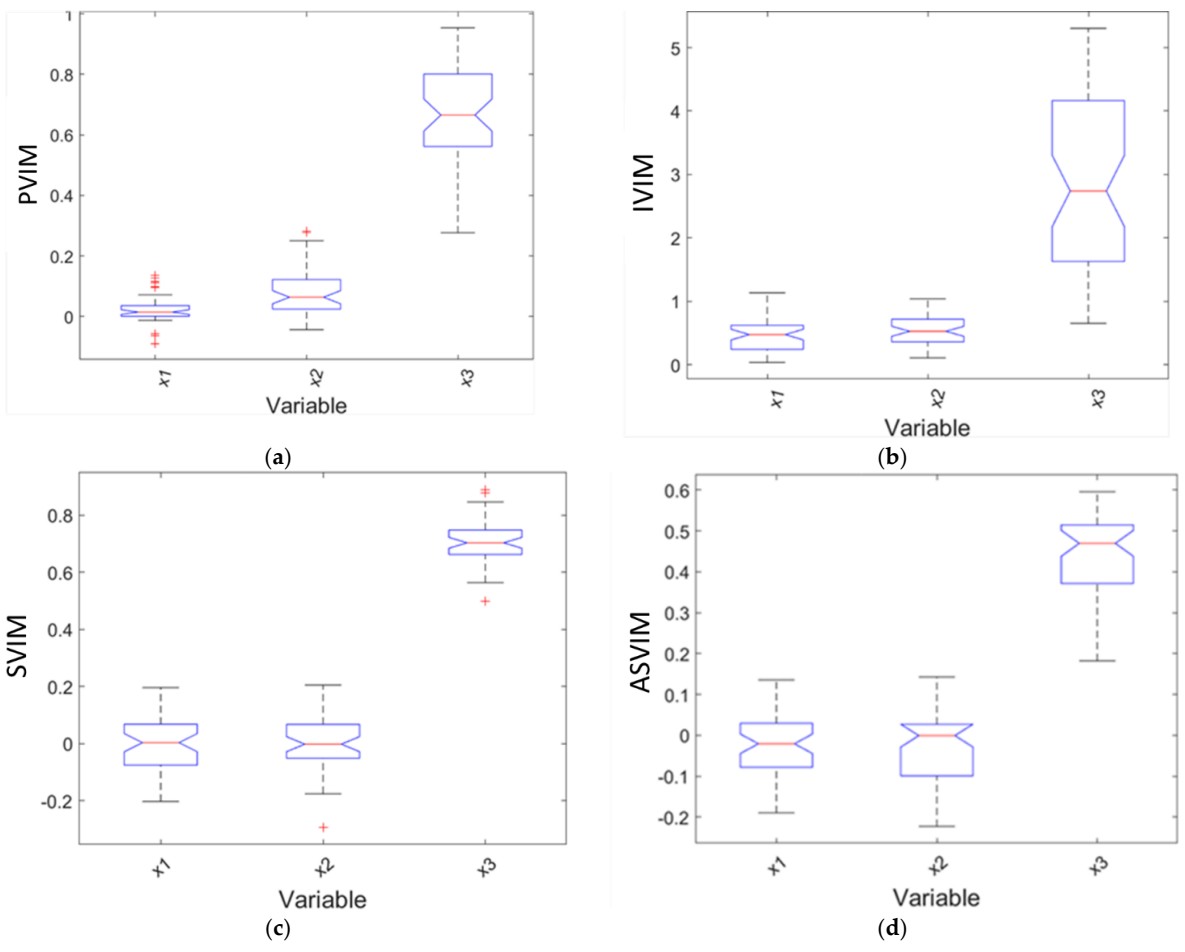

**Figure 3.** Relative importance of the correlated predictors in Case Study 1 based 50 runs, showing (**a**) permutation indices, MSE on OOB data, (**b**) impurity indices on OOB data, (**c**) Shapley value ($R^2$) on test data, and (**d**) approximate Shapley values ($R^2$) on test data.

## 4. Case Study 2: Calcium Carbide Furnace

Historic process data collected daily over an eight-month period of production of $CaC_2$ in an industrial submerged arc furnace previously reported by Aldrich and Reuter [21] were considered in the analysis. These data were also considered by Jemwa and Aldrich [22] in the context of fault diagnosis with kernel-based systems. In contrast to these previous

studies, Shapley value regression was used to facilitate a quantitative analysis of the importance of the operational variables in the furnace.

The 9 predictor variables with 240 samples each are summarized in Table 1. The variables in the last two rows, namely carbide production and carbide grade, were the key performance indicators of the furnace and therefore also the response variables to be predicted from the operational variables.

**Table 1.** Calcium carbide furnace data.

| Variable | Units | Data |
|---|---|---|
| Furnace load | ton | |
| Specific energy consumption | MWh/ton | |
| Electrode resistance | - | |
| Lime consumption | ton/h | |
| Charcoal consumption | ton/h | |
| Coke consumption | ton/h | |
| Anthracite consumption | ton/h | |
| Underburnt lime | % | |
| Overburnt lime | % | |
| Carbide production | ton/h | |
| Carbide grade | L/kg | |

A color-coded image of the correlation matrix of the predictor and target variables is shown in Figure 4. The two target variables showed significant correlation, although the data suggest that this may be attributed to outliers or extreme values at low production rates. The furnace load and resistance also showed significant correlation with carbide recovery. Three predictors in particular were highly correlated: The furnace load and lime consumption (with a binary correlation of R = 0.935), as well as the furnace load and the coke consumption (R = 0.644).

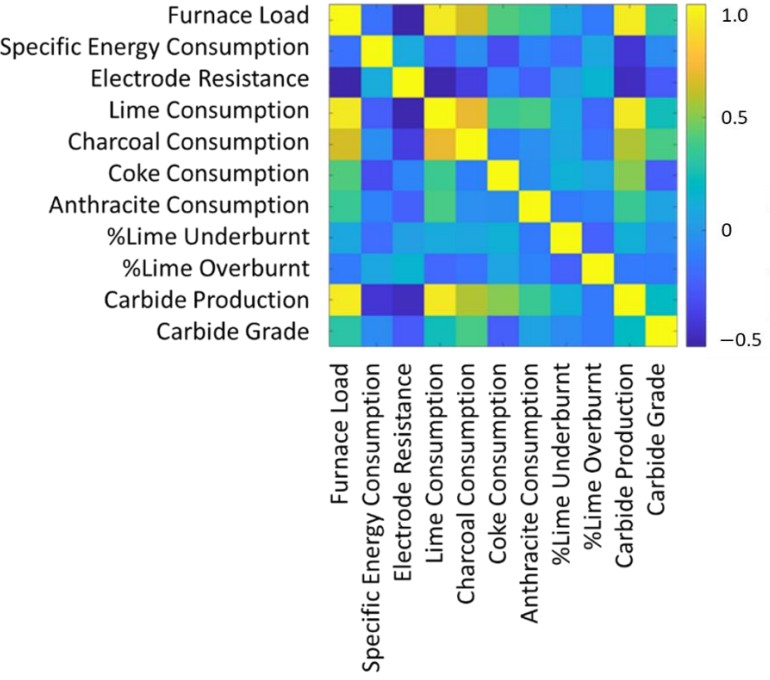

**Figure 4.** Correlogram of the calcium carbide furnace variables.

The multicollinearity of the predictors is summarized by two commonly used indicators, namely the condition index of the predictors, i.e., CI = 70.3, which exceeds the value of 30 that is often used as an indication of significant collinearity [23]. However, based on the variance inflation factors (VIFs) of the numerical predictors summarized in Table 2, and a criterion of >10 [23], only the lime seems to be significantly correlated with the other predictors.

**Table 2.** Variance inflation factors of the calcium carbide furnace predictors.

| Predictor | Furnace Load | Energy | Resistance | Lime | Charcoal | Coke | Anthracite | Lime Under | Lime Over |
|---|---|---|---|---|---|---|---|---|---|
| **VIF** | 9.68 | 1.26 | 1.43 | 14.5 | 5.93 | 3.25 | 2.77 | 1.15 | 1.16 |

Random forest models were fitted to the data, with the following optimal parameters: $K = 100$ $m_{try} = 6$; $n_{try} = 80\%$ of the data; minimum leaf size, $n_{leaf} = 5$. On average, the random forest model could explain 96% of the variance of the response variable. Similar to the previous two case studies, the data were randomly split into training and test sets in an 80:20 ratio for computation of the ASVIMs and SVIMs.

As shown in Figure 5, the four variable importance measures all identified the two most important variables, i.e., the furnace load and lime, but from the third place onwards, results differed. The Shapley measures (SVIM and ASVIM) flagged charcoal as the third most important, significantly explaining approximately 12% of the variance of the target variable. PVIM and IVIM both flagged power consumption as the third most important variable. PVIM and IVIM tended to rank the variables similarly, whereas SVIM and ASVIM provided similar ranking but differing from the predictions of PVIM and IVIM. SVIM and ASVIM yielded the same ranking profiles, but as expected, ASVIM showed larger variance in the results.

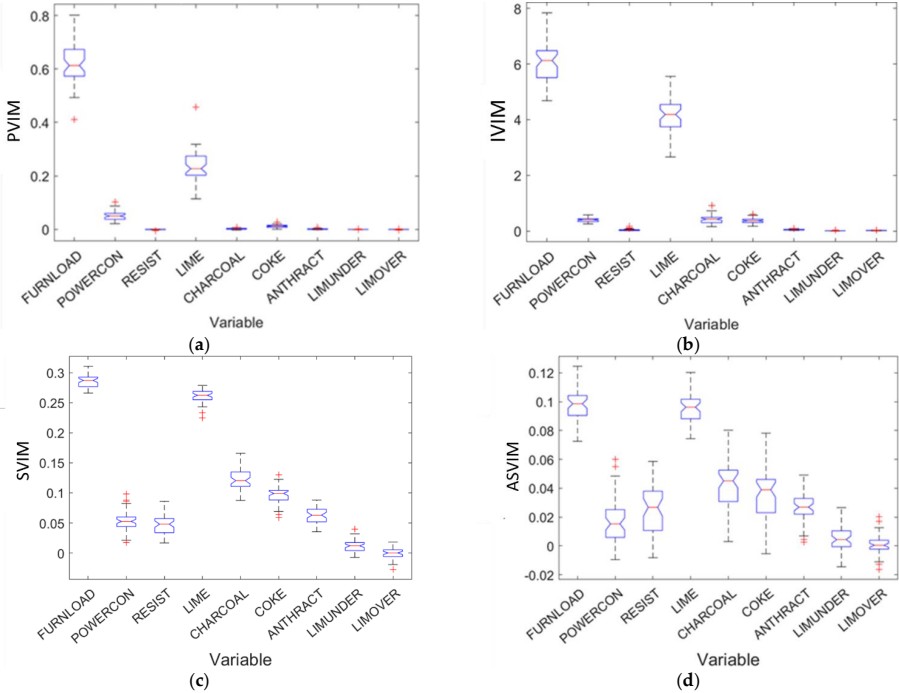

**Figure 5.** Relative importance of predictors in Case Study 2 based on 50 runs, showing (**a**) permutation indices ($R^2$ on test data), (**b**) impurity indices (on OOB data), (**c**) Shapley values ($R^2$ on test data), and (**d**) approximate Shapley values ($R^2$ on test data).

## 5. Case Study 3: Simulation of a Steel Plant

The final case study is based on a public data set that was obtained from the University of California Irvine UCI Machine Learning Repository (UCI Machine Learning Reposi-

tory: Steel Industry Energy Consumption Dataset Data Set). The data set consisted of 35,040 samples of nine input variables, as summarized in Table 3.

**Table 3.** Predictors of energy consumption in a steel plant (Sathishkumar et al. [24]).

| Variable | Symbol | Units |
|---|---|---|
| Energy consumption (target) * | E | kWh |
| Lagging current reactive power ** | LagRP | kVAh |
| Leading current reactive power ** | LeadRP | kVAh |
| Carbon dioxide emission ** | $CO_2$ | ppm |
| Lagging current power factor ** | LagPF | % |
| Leading current power factor ** | LeadPF | % |
| Number of seconds from midnight ** | NSM | s |
| Week status *** | WStatus | 0 (weekend) 1 (weekday) |
| Day of week *** | DWeek | Mon, Tue, Wed, . . . Sun |
| Load type *** | LType | Light, Medium, Maximum |

* Target variable, ** continuous predictor, *** categorical predictor.

Multicollinearity in the predictors could be characterized by a condition index of CI = 296.5, indicative of strong multicollinearity. The variable inflation factors are summarized in Table 4. The $CO_2$ predictor stands out as highly correlated with the other predictors, whereas *LeadRP* and *LeadPF* also show borderline strong multicollinearity.

**Table 4.** Variance inflation factors of the predictors of energy consumption in a steel plant.

| Predictor | LagRP | LeadRP | $CO_2$ | LagPF | LeadPF | NSM |
|---|---|---|---|---|---|---|
| **VIF** | 7.46 | 9.64 | 43.2 | 3.98 | 10.9 | 1.52 |

### 5.1. Boruta Algorithm

Boruta algorithms [25] are wrapper methods built around random forest models.

The Boruta algorithm is initiated by complementing the predictors to be analyzed with shadow predictors, i.e., a permuted version of each variable. A random forest is subsequently fitted to the data, with the original and shadow features as predictors. This model is used to assess the importance of each variable or predictor.

A threshold equal to the performance of the best shadow feature or predictor is calculated, and variables with performance exceeding the threshold are designated as 'hits' on the model. This process is repeated a set number of times, and the hits of all predictors are recorded. The Boruta algorithm does not use a hard limit (% of hits) for retention or elimination of features. Instead, predictors are grouped into three sets based on the binomial distribution of the recorded hits.

### 5.2. Boruta Analysis of Data

The importance of the nine predictors summarized in Table 3 with respect to the energy consumption (*E*) in a steel plant in South Korea was previously investigated by Sathishkumar et al. [24], who made use of a Boruta algorithm for analysis. Their results are summarized in Figure 6, with BVIM indicating the Boruta variable importance measure.

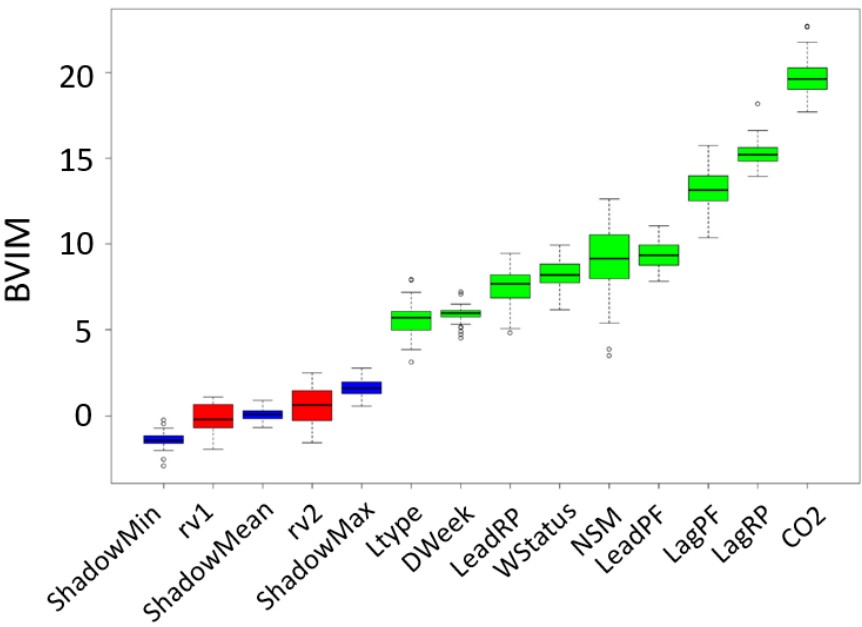

**Figure 6.** Boruta analysis of the importance of predictors of energy use in a steel plant (Sathishikumar et al. [24]).

*5.3. Shapley and Random Forest Variable Importance Measures*

As before, random forest models were fitted to the data, with the following optimal parameters: $m_{try} = 5$; $n_{try} = 80\%$ of the data; minimum leaf size, $m_{leaf} = 5$. On average, the random forest model could explain 99.8% of the variance of the response variable. For the SVIM and ASVIM, the data were randomly divided into a training set and test set with a ratio of 80% to 20%, respectively.

For comparative purposes, the results are visualized in Figure 7. The top row shows the ranking of the variables by the Boruta algorithm, as reported by Sathishkumar et al. [24]. As indicated, all the algorithms were in agreement on the ranking of the first three variables ($CO_2$, *LagRP*, and *LagPF*), except for the random forest with the impurity criterion, which ranked *LType* as the third most important variable.

Predictor ranking are shown in Table 5. The most notable differences are between the random forest using the permutation and impurity criteria and the Boruta algorithm. Boruta (BVIM) ranked *LType* last (ninth position), whereas all the other measures gave ranked it higher (sixth position). This is interesting because the Boruta algorithm is an iterative version of the random forest using the permutation variable importance measure.

**Table 5.** Ranking of predictors of energy consumption in a steel plant based on median values of variable importance measures (BVIM, PVIM, IVIM, SVIM, and ASVIM).

|  | $CO_2$ | *LagRP* | *LagPF* | *LeadRP* | *NSM* | *WStatus* | *LeadPF* | *DWeek* | *LType* |
|---|---|---|---|---|---|---|---|---|---|
| BVIM | 1 | 2 | 3 | 4 | 5 | 6 | 7 | 8 | 9 |
| PVIM | 1 | 2 | 4 | 7 | 6 | 9 | 5 | 8 | 3 |
| IVIM | 1 | 2 | 4 | 7 | 6 | 9 | 5 | 8 | 3 |
| SVIM | 1 | 2 | 3 | 7 | 4 | 9 | 6 | 8 | 5 |
| ASVIM | 1 | 2 | 3 | 7 | 4 | 9 | 6 | 8 | 5 |

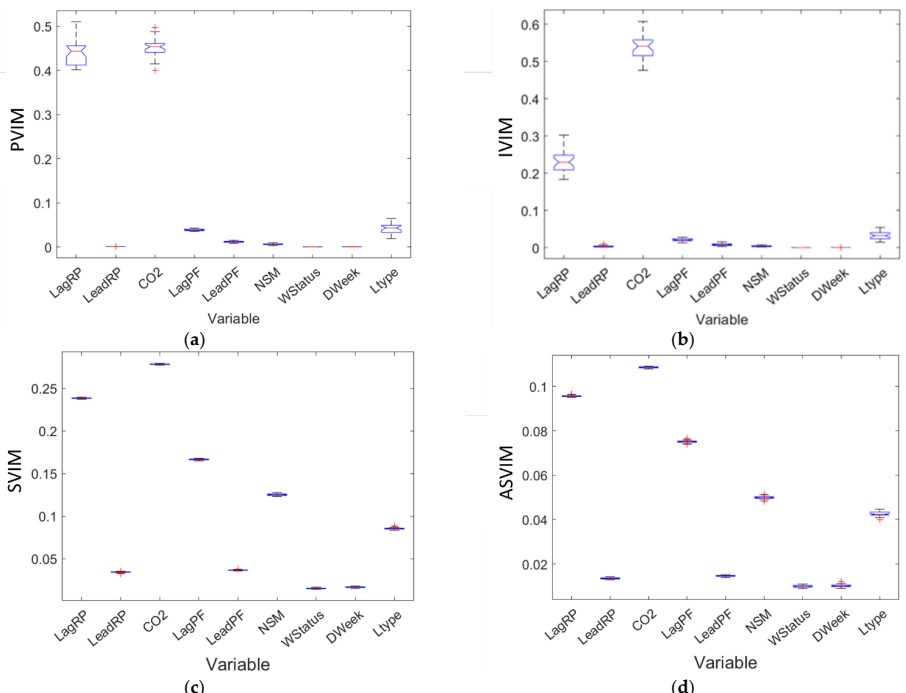

**Figure 7.** Relative importance of predictors in Case Study 3 based on 100 iterations and 30 runs, showing (**a**) permutation indices (MSE on OOB data), (**b**) impurity indices (OOB data), (**c**) Shapley values ($R^2$ on test data), and (**d**) approximate Shapley values ($R^2$ on test data).

The Shapley algorithms and the Boruta algorithm concurred on the ranking of the three most important predictors. In contrast, PVIM and IVIM ranked *LagPF* as the least important, whereas BVIM, SVIM, and ASVIM ranked it in third position. Other differences in the ranking of the variables are also apparent, although it should be borne in mind that the median variable importance measures of the lower ranked predictors are relatively small, and these differences are not as pronounced as those between the top-ranked predictors.

As shown in Table 5, the Shapley algorithm and its approximation yielded identical results. They differ in that the full Shapley algorithm distributes the variance explained by the model (the $R^2$-coefficient) across all the predictors, whereas the approximate Shapley model does not. However, the virtually identical profiles of the variables generated by the two Shapley approaches suggest that the approximated Shapley values can simply be rescaled to similarly yield an estimate of the variance explained by each predictor.

## 6. Discussion of Results

In this investigation, an approximation to Shapley value regression was proposed, and this approach, as well as full Shapley value regression, were empirically compared with other variable importance measures associated with random forests. We found that the approach based on the approximated Shapley value regression yielded results similar to those obtained with the full Shapley value model but at a fraction of the computational cost as a result of linearly scaling with the number of predictors in the model.

Care should be taken when variable importance measures are compared, as the bases for comparison are not the same. For example, in Shapley value models, predictors are evaluated on the basis of their marginal contributions to the model payoff ($R^2$ value in this investigation) in different coalitions of predictor variables (SVIM and ASVIM). The permutation variable importance measure (PVIM) evaluates predictors based on their effect on the model when eliminated from the full model. In contrast, the impurity variable importance measure (IVIM) evaluates the importance of predictors in terms of their inclusion in the structures of the trees of the random forest.

The impurity variable importance measure is popular, owing to its simplicity and speed with which it can be computed. On the downside, impurity variable importance mea-

sures may be biased toward predictors with many possible split points [20,26]. Intuitively, as every possible split point is tested at each node in the tree, continuous or high-cardinality variables will have many more split points to evaluate than categorical or low-cardinality variables. With this multiple testing, there is an increased probability that, purely on a random basis, a predictor may predict the target well and therefore be included in the tree structure of the random forest. This problem is exacerbated when predictors are correlated, as the tree can only consider uniaxial splits. As a consequence, it may underestimate the effect of categorical variables with a few groups only, as indicated by the low ranking of the *LType* predictor in Case Study 3.

In contrast, the permutation variable importance measure is not affected by this bias and is therefore generally preferred over the impurity variable importance measure [27]. However, PVIMs are computationally demanding for high-dimensional data and may provide less robust results than IVIMs [28].

The four variable importance measures (PVIM, IVIM, SVIM, and ASVIM) provided similar results. The approximate Shapley method (ASVIM) provided results similar to those of the Shapley measure (SVIM) but at a fraction of the computational cost. This approach could enable analysis of high-dimensional systems that would not otherwise be possible, owing to computational limitations.

The Boruta variable importance measure considered in Case study 3 has previously been compared with PVIM [29], where it performed better, also outperforming a number of other algorithms. In the last case study in this investigation, it yielded results that differed from those obtained with PVIM, IVIM, SVIM, and ASVIM with regard to the lower-ranked variables.

However, the lower-ranked predictors contributed similarly and weakly to the target variable, which would make exact ranking difficult, as was also suggested by the results of the simulations in Case Study 1.

## 7. Conclusions

Based on the results of this investigation, it can be concluded that full Shapley value models can be reasonably approximated by a combination of their full and single predictor coalitions. This approach yielded predictor rankings similar to those obtained by the full Shapley model. The payoff is a model that scales linearly with the number of predictors to be evaluated, making it applicable to high-dimensional systems, where full Shapley value models cannot be used.

When applied to operational data from a calcium carbide furnace, all variable importance measures identified the furnace load and lime consumption as the most important predictors of throughput. The SVIM and ASVIM measures also indicated the importance of charcoal and coke consumption in the model.

Similar results were obtained with the identification of the most important variables on the consumption of electrical energy in a steel plant previously studied in [24]. The top-three variables for which there was consensus among the Shapley and Boruta variable importance measures explained approximately 67% of the variation in the energy consumption.

**Author Contributions:** Conceptualization, X.L. and C.A.; methodology, X.L. and C.A.; software, X.L.; validation, X.L.; formal analysis, X.L.; investigation, X.L. and C.A.; data curation, X.L. and C.A.; writing—original draft preparation, X.L. and C.A.; writing—review and editing, C.A. and X.L.; project administration, C.A.; funding acquisition, C.A. All authors have read and agreed to the published version of the manuscript.

**Funding:** The authors acknowledge funding support from the Australian Research Council for the ARC Centre of Excellence for Enabling Eco-Efficient Beneficiation of Minerals (Grant Number CE200100009).

**Data Availability Statement:** A publicly available dataset was analyzed in Case Study 3. These data can be found here: http://archive.ics.uci.edu/ml/datasets/Steel+Industry+Energy+Consumption+Dataset.

**Conflicts of Interest:** The authors declare no conflict of interest.

**Nomenclature**

| | |
|---|---|
| $C_i$ | Class to which a sample belongs in a decision tree |
| $CO_2$ | Carbon dioxide emission |
| *COKE* | Coke consumption in a calcium carbide furnace |
| *DWeek* | Day of week |
| $\triangle GI$ | Change in the Gini index that is maximized when the node is split |
| $E$ | Energy consumption |
| $GI_F$ | Gini index of a node in a decision tree |
| $GI_L$ | Gini index of left-descendent node following the split of a parent node in a decision tree |
| $GI_R$ | Gini index of right-descendent node following the split of a parent node in a decision tree |
| $GVIM_i$ | Gini variable importance measure of the $i'$th predictor in a random forest classifier |
| $GVIM_i^{(k)}$ | Gini variable importance measure for the $i'$th predictor in the k'th tree in a random forest classifier |
| $IVIM_i$ | Impurity variable importance measure of the $i'$th predictor in a random forest |
| $l_{max}$ | Maximum depth of a tree in a random forest |
| $K$ | Number of trees in a random forest |
| *LagPF* | Lagging current power factor |
| *LagRP* | Lagging current reactive power |
| *LeadPF* | Leading current power factor |
| *LeadRP* | Leading current reactive power |
| $l_{max}$ | Maximum depth of a tree in a random forest |
| *LType* | Load type |
| $\mathcal{M}$ | Number of predictors in a machine learning model |
| $MSE_k$ | Mean squared error of the $k'$th tree in a random forest |
| $MSE_{k,i}$ | Mean squared error of the $k'$th tree in a random forest when the $i'$th predictor is permuted |
| $m_{try}$ | Number of variables drawn from training data at node splits in a random forest |
| $n$ | Number of samples in a data set |
| *NSM* | Number of seconds from midnight |
| $n_{OOB}$ | Number of OOB samples in a random forest model |
| $n_{leaf}$ | Minimum leaf size in a decision tree |
| $n_{tree}$ | Number of observations drawn from training data at node splits in a random forest |
| $p_F$ | Fraction of samples allocated to a category |
| $p_L$ | Fraction of samples allocated to the left of a split node in a tree |
| $p_R$ | Fraction of samples allocated to the right of a split node in a tree |
| $P(\mathcal{M})$ | Coalition of predictor sets over $\mathcal{M}$ |
| $PVIM_i$ | Permutation variable importance measure of the $i'$th predictor in a random forest |
| $PVIM_i^{(k)}$ | Permutation variable importance measure of the $k'$th tree for the $i'$th predictor in a random forest |
| $R^2$ | Coefficient of determination of a machine learning model |
| $R_v$ | A real number representing the value of a reward function ($v$) |
| $S$ | Coalition of variables or predictors in a machine learning model |
| $SVIM_i$ | Shapley variable importance measure of the $i'$th predictor in a random forest |
| $\mathbf{T}_k$ | OOB data consisting of $n_{OOB}$ samples seen by the $k'$th tree in a random forest during training |
| $\phi_i$ | Shapley value of the $i'$th predictor |
| $v$ | Reward function in Shapley value analysis |
| *WStatus* | Week status |
| $\mathbf{X}$ | Data matrix with $n$ samples of $\mathcal{M}$ predictors |
| $\mathbf{x}_j^{(k)}$ | $i'$th vector of predictors seen by the $k'$th tree in a random forest |
| $Y$ | Labels of variables $X$ |
| $y_j^{(k)}$ | $i'$th label associated with $\mathbf{x}_j^{(k)}$ seen by the $k'$th tree in a random forest |
| $\hat{y}_j^{(k)}$ | $i'$th label estimated by the $k'$th tree in a random forest with input $\mathbf{x}_j^{(k)}$ |
| $\hat{y}_{j\backslash\{i\}}^{(k)}$ | $i'$th label estimated by the $k'$th tree in a random forest with input $\mathbf{x}_j^{(k)}$ not containing the $i'$th predictor |

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
