# Peer review of "Assessing the Influence of Operational Variables on Process Performance in Metallurgical Plants by Use of Shapley Value Regression"

_metals, doi:10.3390/met12111777_

Round 1
Reviewer 1 Report
Hardly any metallurgy or process insight is there in the paper. It uses all kinds of machine learning techniques without giving any process insights. The conclusion drawn is also towards the machine learning technique than the process analyzed.
Author Response
Reviewer 1
Hardly any metallurgy or process insight is there in the paper. It uses all kinds of machine learning techniques without giving any process insights. The conclusion drawn is also towards the machine learning technique than the process analyzed.
Response: The reviewer is correct in that the paper does not focus on metallurgy or process insights as such. While one would ideally like to have considered both the methodology and in-depth insights into metallurgical systems that could be uncovered by application of the methodology, in practice one of the two themes have to take precedence.
In this investigation, the focus was on analytical methodologies used by metallurgical engineers and practitioners to gain insights into metallurgical systems. This includes a novel variant of the Shapley approach that can be used with systems, where many variables need to be assessed and also highlight issues that need to be considered when machine learning models are used to interpret data from metallurgical systems.

Reviewer 2 Report
The method presented by the authors is a simplified alternative to a statistical-mathematical dispersion analysis of the influence factors in a complex process with many input variables and of the significant influence intervals of the most influential process parameters. The results are presented explicitly, the diagrams, figures, tables are sufficient to support the authors conclusions.
The paper presents 29 bibliographical references, sufficient considering the nature of the research.
The similarity test gave a result of 17%, 5% representing taking over without citing some data from the work of one of the two authors, namely" Process Variable Importance Analysis by Use of Random Forests in a Shapley Regression Framework", published in Minerals 2020, 10(5), 420: https://doi.org/10.3390/min10050420.
The value given by the similarity test was acceptable. I consider that the paper can be published in the form presented.

Author Response
Reviewer 2
The paper presented by the authors is interesting for the scientific community.
In this work, the authors present a diagnostic model of the development of metallurgical processes based on the regression analysis of values using the Shapley method.
Three case studies are presented, from different environments to which this method can be applied.
The authors demonstrate that in certain situations the approach based on the approximate regression of the Shapley value gives results like the complete Shapley model.
The method presented by the authors is a simplified alternative to a statistical-mathematical dispersion analysis of the influence factors in a complex process with many input variables and of the significant influence intervals of the most influential process parameters. The results are presented explicitly, the diagrams, figures, tables are sufficient to support the authors conclusions.
The paper presents 29 bibliographical references, sufficient considering the nature of the research.
The similarity test gave a result of 17%, 5% representing taking over without citing some data from the work of one of the two authors, namely" Process Variable Importance Analysis by Use of Random Forests in a Shapley Regression Framework", published in Minerals 2020, 10(5), 420: https://doi.org/10.3390/min10050420.
The value given by the similarity test was acceptable. I consider that the paper can be published in the form presented.
Response: The authors appreciate the positive comments from the reviewer.

Reviewer 3 Report
Dear authors,
Some lines in the text are in blue color and this is somethong to be fixed for proofing the manuscript.
Author Response
Reviewer 3
Dear authors,
Some lines in the text are in blue color and this is something to be fixed for proofing the manuscript.
Response: All text in the revised paper has been revised to ensure that it is uniformly black in colour.
Reviewer 4 Report
Dear authors!
Your manuscript is very interesting. I think it expands our ability to apply machine learning in industrial processes. Some comments I placed below:
1) Why did you use the random forest and Shapley value regression algorithms only? Why did you not use neural networks or other algorithms? How you performed data preprocessing? You should indicate it in detail.
2) How you divided data on train and test sets?
3) Line 198-199: You should add the link of dataset
4) Line 198-199: Why did you use as targets only carbide production and carbide grade? The specific energy consumption also important indicator for furnace operation
5) I do not see comparison the accuracy of models for random forest and Shapley value regression for targets.
Author Response
Reviewer 4
Dear authors!
Your manuscript is very interesting. I think it expands our ability to apply machine learning in industrial processes. Some comments I placed below:
1) Why did you use the random forest and Shapley value regression algorithms only? Why did you not use neural networks or other algorithms? How you performed data preprocessing? You should indicate it in detail.
Response: No doubt, neural networks are important analytical tools, but they have not included in the present study, as this would have required substantial discussion on the different methods to interpret neural networks, for example based on analysis of the weights of the connection paths between inputs and outputs or making use of sensitivity and perturbation analysis. Also, ideally one would have to consider the effect of different neural network architectures, so the authors intend to use neural networks in a separate study in future work, where these methods would be considered systematically.
2) How you divided data on train and test sets?
Response: The data were randomly divided into training and test sets by a ratio of 80 to 20, for the calculation of SVIM and ASVIM only. For the other two methods based on the Gini and permutation criteria, the out-of-bag results were used and the This has been highlighted in the revised manuscript in each case study with a sentence specifically pointing out that for the SVIM and ASVIM, the data were randomly divided into training and test sets in a respective ratio of 80 to 20.
3) Line 198-199: You should add the link of dataset
Response: Unfortunately, a link to the dataset used in Case Study 2 cannot be provided, as it is not a public dataset.
4) Line 198-199: Why did you use as targets only carbide production and carbide grade? The specific energy consumption also important indicator for furnace operation
Response: That is a valid point, but as the specific energy consumption and the carbide grade and throughput are interrelated, the authors have decided to focus on the carbide production and grade only.
5) I do not see comparison the accuracy of models for random forest and Shapley value regression for targets.
Response: In each case study, the accuracies of the different models is similar as the Shapley values were obtained from a collection of R2 scores of the random forest models.

Reviewer 5 Report
The paper titled “Assessing the Influence of Operational Variables in Metallurgical Plants on Process Performance by Use of Shapley Value Regression” reports the application of Shapley regression analysis with random forest models used to evaluate the influence of different predictors on the performance of models based on simulated data, data from a calcium carbide furnace, as well as data from a steel plant. The paper is excellent in quality. I recommend it for publication after major revision. The comments are listed below.
1. The reference style in the text is out of requirements.
2. The formatting of tables, figure and table captions are out of requirements.
3. Figures 2, 3, 5, 7: The plots should be labeled as (a), (b)… The captions should be modified accordingly.
4. List of Symbols (one more section or Appendix) should be added. All the symbols used in the text should be defined there.
5. Figure 6: What is Shaowmax?
6. Pages 11-12: The font color looks strange.
Author Response
Reviewer 5
The paper titled “Assessing the Influence of Operational Variables in Metallurgical Plants on Process Performance by Use of Shapley Value Regression” reports the application of Shapley regression analysis with random forest models used to evaluate the influence of different predictors on the performance of models based on simulated data, data from a calcium carbide furnace, as well as data from a steel plant. The paper is excellent in quality. I recommend it for publication after major revision. The comments are listed below.
1. The reference style in the text is out of requirements.
Response: The reference style in the text has been changed to a numbered format in the revised manuscript.
2. The formatting of tables, figure and table captions are out of requirements.
Response: The formatting of tables, figure and table captions have been revised accordingly.
3. Figures 2, 3, 5, 7: The plots should be labeled as (a), (b)… The captions should be modified accordingly.
Response: The labels and captions of the subplots of Figures 2, 3, 5, 7, as highlighted in the revised manuscript.
4. List of Symbols (one more section or Appendix) should be added. All the symbols used in the text should be defined there.
Response: A table of symbols and their definitions has been added as an appendix in the revised manuscript.
5. Figure 6: What is Shaowmax?
Response: The typographical error has been corrected, to read “ShadowMax” in the revised paper.
6. Pages 11-12: The font color looks strange.
Response: Thanks for pointing this out. The font color of the text has been changed to black to be consistent with the rest of the text.

Round 2
Reviewer 5 Report
The authors have revised the manuscript. However, minor revision is still required. The comments are listed below.
1. Appendix A should be placed before references (see template).
2. Article numbers should be added in references 2, 3, 5, 6, 8, and 18.
3. I have studied the profile of Sathishkumar (https://orcid.org/0000-0002-8271-2022). So, I guess that ref. 24 should be Sathishkumar, V.E.; Shin, C.; Cho, Y. Moreover, the authors also wrote “as reported by Sathishkumar 295 et al. [24].”